# Developmental outcomes of an individualised complementary feeding intervention for stunted children: a substudy from a larger randomised controlled trial in Guatemala

Boris Martinez,[1,2] Sayra Cardona,[3] Patricia Rodas,[3] Meri Lubina,[3] Ana Gonzalez,[3] Meghan Farley Webb,[1,4] Maria del Pilar Grazioso,[3] Peter Rohloff[1,5]

► Additional material is published online only. To view, please visit the journal online (http://dx.doi.org/10.1136/bmjpo-2018-000314).

[1]Wuqu' Kawoq, Maya Health Alliance, Santiago Sacatepéquez, Guatemala
[2]Department of Medicine, Saint Peter's University Hospital, New Brunswick, New Jersey, USA
[3]Department of Psychology, Universidad del Valle de Guatemala, Guatemala City, Guatemala
[4]Department of Anthropology and Sociology, Albion College, Albion, Michigan, USA
[5]Division of Global Health Equity, Brigham and Women's Hospital, Boston, Massachusetts, USA

**Correspondence to**
Dr Peter Rohloff; prohloff@bwh.harvard.edu

## ABSTRACT

**Objective** Stunting is a common cause of early child developmental delay; Guatemala has the fourth highest rate of stunting globally. The goal of this study was to examine the impact of an intensive community health worker-led complementary feeding intervention on early child development in Guatemala. We hypothesised that the intervention would improve child development over usual care.

**Design** A substudy from a larger individually randomised (1:1 allocation ratio), parallel-group superiority trial, with blinding of study staff collecting outcomes data.

**Setting** Rural, indigenous Maya communities in Guatemala.

**Participants** 210 stunted children (height-for-age z-score ≤−2.5) aged 6–24 months, previously randomised to usual care (106) or an intensive complementary feeding intervention (104). 84 in the intervention and 91 in the usual care arm agreed to participate.

**Interventions** Community health workers conducted monthly home visits for 6 months, providing usual care or individualised complementary feeding education.

**Main outcome measures** The primary outcomes were change in z-scores for the subscales of the Bayley Scales of Infant Development (BSID), Third Edition.

**Results** 100 individuals were included in the final analysis, 47 in the intervention and 53 in the usual care arm. No statistically significant differences in age-adjusted scores between the arms were observed for any subscale. However, improvements within-subjects in both arms were observed (median duration between measurements 189 days (IQR 182–189)). Mean change for subscales was 0.45 (95% CI 0.23 to 0.67) z-scores in the intervention, and 0.43 (95% CI 0.25 to 0.61) in the usual care arm.

**Conclusions** An intensive complementary feeding intervention did not significantly improve developmental outcomes more than usual care in stunted, indigenous Guatemalan children. However, both interventions had significant positive impacts on developmental outcomes.

**Trial registration number** NCT02509936.

**Stage** Results.

### What is already known on this topic?

► Stunting is the most common cause of early childhood developmental delays, and nearly half of Guatemalan children are stunted, with higher prevalence among the Guatemalan indigenous Maya population.
► Stunting early in life has deleterious long-term impacts on educational attainment, intellectual outcomes and adult economic earning potential. Few investigations of these effects exist for the Guatemalan indigenous Maya population.
► Complementary feeding practices interventions are the cornerstone for stunting prevention and treatment; however, investigation of the effects of these interventions on developmental outcomes are limited.

### What this study hopes to add?

► Both standard and augmented behaviour change approaches to improved complementary feeding practices by caregivers have important positive impacts on the development of stunted children.
► Current public malnutrition reduction strategies being implemented in indigenous Maya communities in Guatemala can improve childhood developmental outcomes.
► Developmental assessments can be feasibly adapted and contextualised to a rural, indigenous population at risk.

## INTRODUCTION

In low-income and middle-income countries, 43% of children under 5 are at risk of not reaching their developmental potential, and 70% of the attributable risk for delays in early childhood development (ECD) is due to stunting, or low height-for-age.[1 2] Guatemala, a Central American country of 17 million

inhabitants, has the highest rate of stunting in the Western Hemisphere.[3] The burden of stunting disproportionately affects the rural, indigenous Maya population, where stunting often exceeds 70% and complementary feeding and dietary diversity indicators are poor.[4–6]

Early scientific research in Guatemala was critical to the international community's understanding of stunting's impact on human capital. The Institute of Nutrition of Central America and Panama cohort study—conducted from 1969 to 1977, with long-term follow-up—demonstrated the deleterious impacts of stunting on educational attainment, intellectual outcomes and adult economic earning potential.[7] Subsequently, there have been extensive investigations of stunting's nutritional and sociodemographic correlates in the indigenous Maya population. However, there have been few investigations of developmental outcomes or of the impact of contemporary nutritional interventions on ECD.[8–11]

Given the gaps in adequate complementary feeding practices and diet quality, interventions to promote caregiver knowledge around infant-child feeding are an important strategy in Guatemala.[5 6 12] This strategy is supported by international evidence, with several studies showing that linear growth is strongly correlated with better development and that complementary feeding education interventions improve stunting.[11 13–15] However, there has been limited direct investigation of the effect of complementary feeding on developmental outcomes.[16–18] This is an important knowledge gap, because feeding interventions may at times have small direct impacts on linear growth velocity but conceivably larger impacts on developmental outcomes. Therefore, directly measuring these developmental outcomes may generate additional evidence supporting the interventions, which could have important policy implications in a country like Guatemala, where debate over the value of standard child nutrition programme offerings is ongoing.[19 20]

In this study, we contribute by assessing developmental outcomes in indigenous Maya infants in rural Guatemala. We hypothesised that an individualised, intensive approach to caregiver complementary feeding education would improve outcomes over usual care. We individually randomised child-caregiver dyads (aged 6–24 months, height-for-age z-score ≤−2.5) to 6 months of usual care, which included home-based growth monitoring and micronutrient and food supplementation, versus usual care augmented with individualised complementary feeding education, using monthly structured interviews and active goal-setting to promote incremental improvements.

## METHODS
### Study context
This study was conducted at Maya Health Alliance (MHA), a primary healthcare organisation working in rural Maya communities. At MHA, nutrition community health workers (CHW) provide home-based services to children aged 6–24 months with stunting.[20] The study was conducted in Tecpán, Chimaltenango (population 95 000), with a settlement of agricultural Kaqchikel Maya families living 25 km from the town centre. The study was conducted according to the principles of Declaration of Helsinki.

### Trial design
This was a planned substudy on developmental outcomes within a larger, individually randomised, two-arm trial comparing individualised complementary feeding caregiver education with usual care (Clinicaltrials.gov Identifier: NCT02509936), described in detail elsewhere.[20] Briefly, eligibility criteria included: children aged between 6 and 24 months and length-for-age z-score ≤−2.5 SD by WHO standards.[21] Exclusion criteria were acute malnutrition (weight-for-length z-score of ≤−2 SD) or severe medical illness. Study interventions were delivered by two CHW teams, each consisting of two CHWs. One team provided usual care while the other provided the intervention. Child-caregiver dyads were recruited and written informed consent obtained by a study staff member not involved in the intervention. At enrolment, child's anthropometric and dietary feeding practices as well as household demographic and socioeconomic characteristics were obtained. Household developmental stimulation data was gathered at enrolment using the Family Care Indicators (FCI) interview.[22] Due to the considerable time and expense constraints related to administering psychometric testing, a subset of subjects, namely those consecutively recruited for the larger study during the first 5 months (n=210, figure 1), were invited to participate in the substudy. Study outcomes (subscale scores on the Bayley Scales of Infant Development, Third Edition (BSID-III))[23] were obtained at 0 and 6 months by trained study psychologists from the Universidad del Valle de Guatemala, supported by MHA interpreters (Maya Kaqchikel/Spanish). Both psychologists and interpreters were blinded to study allocation.

### Study interventions
The study duration was 6 months. As described elsewhere,[24] the usual care arm received a monthly home-visit regimen from a team of CHWs including growth monitoring, daily multiple micronutrient powder supplement (ferrous fumarate 12.5 mg, zinc gluconate 5 mg, retinol acetate 300 μg, folic acid 160 μg and ascorbic acid 30 mg; Prodipa, Guatemala City, Guatemala); and a biweekly food ration (beans 1000 g, eggs 20 units and *Incaparina* 900 g (a common corn and soy-based food supplement; Alimentos, Guatemala City, Guatemala)). For the intervention arm, subjects received usual care and monthly individualised caregiver counselling focused on improving meal frequency and dietary diversity for the child,[25] from a separate CHW team.

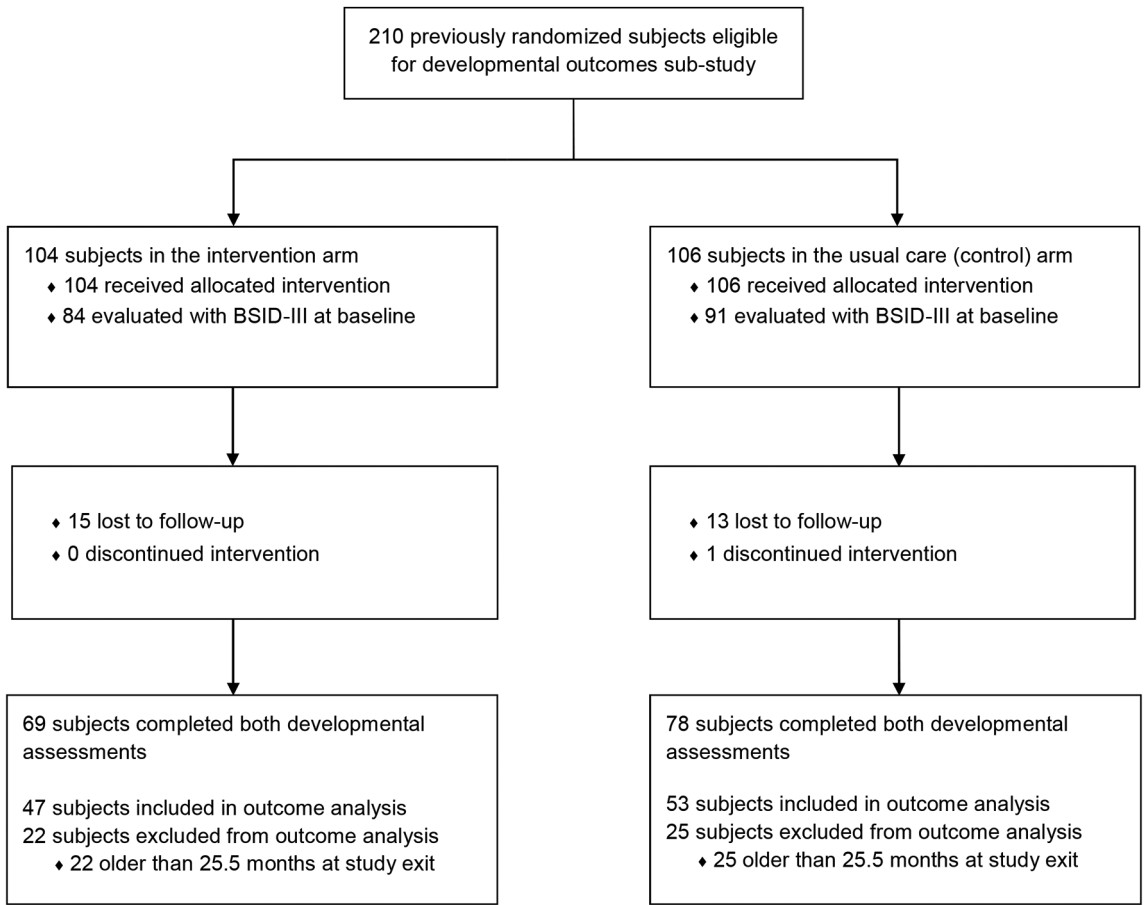

**Figure 1** Subject enrolment, randomisation and follow-up. BSID-III, Bayley Scales of Infant Development, Third Edition.

### Study outcomes and measure

The primary outcome was a change in z-score for subscales of the BSID-III, calculated as described below. In preparation for use of the BSID-III, our team conducted a review of prior studies of psychometric testing with Guatemalan children, with the majority using previous versions of the BSID.[9–11 26] Despite its technical complexity, the BSID is widely used as a gold-standard instrument worldwide.[27 28] Additionally, our lead psychologist (MPG) had extensive prior experience successfully using the tool in Guatemala. For these reasons, the BSID-III was chosen.

In preparation for testing, our team (interpreters, linguists, psychologists, anthropologists, nurses and physicians) reviewed existing BSID-III English and Spanish materials.[23 29] Modifications for idiomatic Guatemalan Spanish and translations into Kaqchikel Maya were produced through group discussion and audience testing with volunteer caregivers and health workers. Forward translation, back translation, harmonisation and cognitive debriefing were used to ensure accuracy and cultural adaptation. A visual Likert scale assisted with administration of socioemotional scales (see online supplementary figure 1). A small number of vocabulary and visual stimulus items were substituted to be culturally or context appropriate (eg, sedan car substituted for pick-up truck).

Nine psychologists (bilingual Spanish/English; three postgraduate and six graduate or undergraduate) who had formal training in infant/child assessments as part of their education participated in a 1-week long training in rural data collection and multicultural competencies, including active BSID-III practice sessions. Five Maya interpreters (Kaqchikel Maya/Spanish speaking; the majority with a high school teaching degree) participated in a 1-week training with active BSID-III practice sessions. Senior study psychologist (MPG), with significant professional experience in infant/child assessments, supervised BSID-III training and data collection procedures. Testing was performed by a team of one psychologist and one interpreter, supervised by MPG and the principal study coordinator (BM).

### Sample size and randomisation

For the primary outcome, we calculated that enrolment of 72 subjects per arm would detect a change of 0.5 SD on the BSID-III composite score, with an α of 0.05, power of 80% and 15% lost to follow-up. All subjects recruited for the larger study were allocated by simple randomisation from a computer-generated list of random numbers, no further randomisation or allocation were performed for this substudy.

### Statistical analyses

Descriptive statistics were calculated using Stata V.13. Family poverty scores (possible score range: 0–100) were calculated as described.[30] Overall FCI scores were calculated as the sum of 18 item scores.[22] Differences between arms were assessed using the Student's t-test or Wilcoxon-Mann-Whitney U test for continuous variables and the $X^2$ or Fisher's exact tests for categorical variables. Analysis was by intention-to-treat, except where loss to follow-up prevented collecting outcome data or baseline age-band BSID-III data were unavailable to calculate exit BSID-III z-scores.

Raw scores for each BSID-III subscale (cognitive, receptive and expressive language, fine and gross motor, socioemotional) were calculated as the sum of assessment items performed. BSID-III norms do not exist for Guatemala, and there are concerns about the validity of applying BSID-III norms from a different population. This is the case because, for example, some test items are most appropriate for urban, literate populations (eg, use of books in the testing process) and because some language items require adaptation to the syntactic structure of Mayan languages.[31 32] Furthermore, after data collection for the study was completed, interim ethics board review requested that we not use external norms in our analysis plan. Therefore, study-internal z-scores for each BSID-III subscale were calculated based on the raw scores distribution of each age-band during baseline measurements, an approach used in other studies.[33 34] These were then used to calculate individual subject z-scores for both 0 and 6 month timepoints. Since all subjects recruited were younger than 24 months, baseline BSID-III z-score data distribution was unavailable for subjects in older age bands (>25.5 months) at study exit, hence subjects older than 25.5 months were excluded from the primary outcome analysis.

**Table 1**  Baseline demographic, clinical and BSID-III characteristics of study participants

| Characteristic* | Individualised education (intervention) arm (n=84) | Usual care arm (n=91) | P values† |
|---|---|---|---|
| Maternal characteristics | | | |
| Age (years) | 26.08±6.84 | 27.15±6.59 | 0.29 |
| Education (years) | 2 (0–3.5) | 2 (0–4) | 0.93 |
| Literacy, no. (%) | 44 (51.65) | 47 (52.38) | 0.92 |
| Parity | 2.5 (1–5) | 3 (2–5) | 0.43 |
| Child characteristics | | | |
| Male, no. (%) | 49 (58.33) | 57 (62.64) | 0.56 |
| Age at BSID-III evaluation (months) | 16.32±4.93 | 15.84±5.21 | 0.53 |
| Height-for-age z-score | −3.40±0.69 | −3.41±0.73 | 0.91 |
| Weight-for-age z-score | −2.01±0.74 | −1.94±0.82 | 0.54 |
| Weight-for-length z-score | −0.26±0.86 | −0.13±0.90 | 0.35 |
| Feeding practices indicators | | | |
| Minimum dietary diversity, no. (%) | 59 (70.24) | 49 (53.85) | 0.03 |
| Minimum meal frequency, no. (%) | 68 (80.95) | 81 (89.01) | 0.13 |
| Minimum acceptable diet, no. (%) | 51 (60.71) | 43 (47.25) | 0.07 |
| Household characteristics | | | |
| No. of children under 5 years | 2 (1–2) | 2 (1–2) | 0.18 |
| Family poverty score | 28.33±11.64 | 27.35±9.17 | 0.53 |
| Family care indicators score | 8.59±2.63 | 8.47±2.15 | 0.74 |
| BSID-III subscales z-scores‡ | | | |
| Cognitive | −0.05±1.00 | 0.04±0.97 | 0.53 |
| Receptive language | −0.02±0.96 | 0.02±1.00 | 0.81 |
| Expressive language | 0.01±0.94 | −0.01±1.02 | 0.88 |
| Fine motor | −0.15±0.94 | 0.14±1.00 | 0.06 |
| Gross motor | −0.10±0.91 | 0.10±1.04 | 0.18 |
| Socioemotional | 0.05±1.15 | −0.02±0.95 | 0.65 |

*Data presented as means±SD, median (IQR) or no. (%), as appropriate.
†Student's t-test or Wilcoxon-Mann-Whitney U test for continuous variables, $X^2$ or Fisher's exact tests for categorical variables, as appropriate.
‡ BSID-III, Bayley Scales of Infant Development, Third Edition.

We conducted an adjusted exploratory analysis to investigate the effect of prespecified covariates on the primary outcomes, including maternal parity and education, gender, number of under-5 children in the home, household socioeconomic status and age and length-for-age at enrolment. We constructed a hierarchical linear model for BSID-III subscales (MIXED function, Stata V.13). Non-significant covariates were removed from the model using serial likelihood ratio tests.[35] The best-fit model for all BSID-III subscales was chosen using the Breusch-Pagan Langrange multiplier test.

## RESULTS

### Participants

Eligible participants were recruited in rolling fashion from August to December 2015. Final participants exited in July 2016. A total of 210 children were eligible for the substudy (104 intervention arm, 106 usual care arm). Eighty-four children in the intervention and 91 in the control arm received BSID-III evaluation at study entry. Baseline characteristics of participants in the two study arms were well balanced, except for children in the intervention arm having greater minimum dietary diversity (table 1).

The cumulative incidence of loss to follow-up was 18% (n=15) in the intervention and 14% (n=13) in the usual care arm (figure 1). One child in the control arm discontinued treatment. Furthermore, 22 children in the intervention arm and 25 in the control arm with BSID-III data at study exit were older than 25.5 months. Baseline z-score data distribution was not available for this older age band. Online supplementary table 1 compares the characteristics of subjects who completed the study and both BSID-III evaluations (n=147) with those lost to follow-up or who did not complete both assessments (n=63). There were no statistically significant differences for major characteristics. Online supplementary table 2 compares the baseline characteristics of subjects excluded from the outcomes analysis due to age at study exit with those included. Subjects excluded were similar to those included, except for having lower minimum meal frequency (72% for excluded vs 88% for included subjects) and being older (21.97±1.71 vs 13.60±3.95 months). Difference in age was an expected finding, since our analysis was restricted to subjects younger than 25.5 months at study end point, since baseline BSID-III data distribution for z-scores calculation was unavailable for the older age bands.

### Outcomes

The analysis of primary outcomes was by intention-to-treat for subjects younger than 25.5 months at study exit in this substudy, including one child who discontinued treatment. One hundred subjects were included in the primary analysis (47 intervention arm, 53 usual care arm; figure 1). For developmental outcomes (table 2), we observed positive changes in most BSID-III subscales z-scores over the 6-month period in both study arms (median duration between measurements 189 days (IQR 182–189)). Mean change for subscales was 0.45 (95% CI 0.23 to 0.67) z-scores in the intervention arm, and 0.43 (95% CI 0.25 to 0.61) z-scores in the usual care arm. No statistically significant differences were observed between the two study arms.

### Exploratory analyses

We used a hierarchical linear regression model to estimate changes in each BSID-III subscale z-score as a function of prespecified covariates, while controlling for subject-level variation over time. Our final model included gender, age at enrolment, maternal parity and school attendance, number of children under 5 in the household and study arm. The adjusted analysis was consistent with our unadjusted primary analysis, with statistically significant improvements in developmental outcomes over the study period in all subscales (table 3), despite no difference between the study arms. Maternal school attendance was associated with greater positive expressive language and gross motor developmental changes (change at 6 months of 0.34, 95% CI 0.06 to 0.62, and 0.31, 95% CI 0.04 to 0.59 z-scores, respectively). Improvements in the cognitive subscale z-scores were more pronounced for males (0.28, 95% CI 0.03 to 0.53).

**Table 2** Key developmental outcomes

| z-Scores change for BSID-III subscales*‡ | Individualised education (intervention) arm (n=47) | Usual care arm (n=53) | Difference |
|---|---|---|---|
| Cognitive | 0.28 (–0.14 to 0.71) | 0.38 (0.05 to 0.72) | –0.10 (–0.63 to 0.43) |
| Receptive language | 0.49 (0.11 to 0.86) | 0.56 (0.21 to 0.92) | –0.08 (–0.58 to 0.43) |
| Expressive language | 0.69 (0.38 to 0.99) | 0.63 (0.34 to 0.93) | 0.05 (–0.36 to 0.48) |
| Fine motor | 0.40 (0.04 to 0.76) | 0.27 (–0.09 to 0.63) | 0.13 (–0.38 to 0.63) |
| Gross motor | 0.70 (0.30 to 1.10) | 0.51 (0.18 to 0.85) | 0.19 (–0.32 to 0.70) |
| Socioemotional | 0.20 (–0.13 to 0.53) | 0.44 (0.07 to 0.81) | –0.24 (–0.73 to 0.25) |
| Mean change | 0.45 (0.23 to 0.67) | 0.43 (0.25 to 0.61) | 0.02 (–0.25 to 0.30) |

*Values are mean z-score change (95% CI).
‡BSID-III, Bayley Scales of Infant Development, Third Edition.

**Table 3** Estimates and 95% CI from hierarchical linear mixed models for z-scores change in BSID-III subscales‡

| Covariates | Cognitive (n=176) | Receptive language (n=176) | Expressive language (n=174) | Fine motor (n=174) | Gross motor (n=175) | Socioemotional (n=185) |
|---|---|---|---|---|---|---|
| Adjusted z-score change from study entry to exit | 0.41* (0.09 to 0.73) | 0.50** (0.18 to 0.81) | 0.69** (0.42 to 0.96) | 0.32* (0.01 to 0.63) | 0.58** (0.27 to 0.89) | 0.52** (0.22 to 0.82) |
| Gender (0=female, 1=male) | 0.28* (0.03 to 0.53) | −0.07 (−0.32 to 0.18) | −0.08 (−0.33 to 0.18) | 0.03 (−0.21 to 0.26) | −0.05 (−0.30 to 0.21) | −0.06 (−0.32 to 0.19) |
| Age at enrolment (months) | | | | | | |
| 6–12 | – | – | – | – | – | – |
| 13–18 | 0.08 (−0.23 to 0.39) | 0.06 (−0.24 to 0.37) | −0.13 (−0.45 to 0.18) | 0.13 (−0.16 to 0.42) | 0.07 (−0.24 to 0.38) | −0.09 (−0.41 to 0.24) |
| 19–24 | 0.22 (−0.10 to 0.53) | 0.07 (−0.24 to 0.38) | 0.06 (−0.26 to 0.37) | 0.20 (−0.10 to 0.50) | 0.20 (−0.11 to 0.52) | −0.15 (−0.48 to 0.17) |
| Maternal parity (≤2, ≥3) | −0.26 (−0.05 to 0.51) | −0.02 (−0.32 to 0.28) | −0.09 (−0.39 to 0.22) | −0.14 (−0.43 to 0.15) | −0.09 (−0.40 to 0.21) | −0.25 (−0.54 to 0.05) |
| Maternal school attendance (none, some) | −0.02 (−0.30 to 0.26) | 0.10 (−0.17 to 0.38) | 0.34* (0.06 to 0.62) | 0.14 (−0.12 to 0.40) | 0.31* (0.04 to 0.59) | −0.05 (−0.33 to 0.24) |
| No. of children under 5 | | | | | | |
| 1 | – | – | – | – | – | – |
| 2 | 0.23 (−0.05 to 0.51) | 0.02 (−0.26 to 0.30) | 0.00 (−0.28 to 0.28) | 0.14 (−0.12 to 0.41) | 0.02 (−0.26 to 0.30) | −0.25 (−0.54 to 0.05) |
| ≥3 | 0.36 (−0.10 to 0.82) | −0.13 (−0.59 to 0.32) | 0.19 (−0.27 to 0.66) | 0.08 (−0.36 to 0.51) | 0.28 (−0.18 to 0.74) | −0.16 (−0.63 to 0.32) |
| Study arm (0=usual care, 1=intervention) | −0.08 (−0.38 to 0.21) | −0.04 (−0.32 to 0.25) | 0.01 (−0.27 to 0.29) | −0.27 (−0.55 to 0.00) | −0.20 (−0.49 to 0.09) | −0.01 (−0.30 to 0.28) |
| Likelihood Ratio test | 0.11 | 0.13 | 0.0008 | 0.20 | 0.05 | 0.007 |

*p<0.05, **p<0.01.
‡BSID -III, Bayley Scales of Infant Development, Third Edition.

## DISCUSSION

In this paper, we describe developmental outcomes from a substudy of a larger individually randomised clinical trial of a complementary feeding intervention in a rural indigenous area of Guatemala with high stunting prevalence. We found significant improvements across multiple developmental subscales over the study period for children in both the usual care (mean change of 0.43 (95% CI 0.25 to 0.61) z-scores) and the intervention (mean change of 0.45 (95% CI 0.23 to 0.67) z-scores) arms. These improvements remained highly statistically significant after controlling for important covariates (table 3) and occurred despite the larger study showing only non-significant improvements in linear growth, as previously reported.[24]

No statistically significant difference in improvements was observed between the study arms, suggesting that both usual care and intensive approaches were equally effective in promoting development. Important limitations of our study lead to two alternative explanations for this finding. First, when designing the trial, usual care was to be delivered by an existing public sector rural programme. However, widespread closures of these public services happened during study preparation.[19] Therefore, MHA's CHWs agreed to institute the usual care arm. Since these CHWs conduct activities using home visits (rather than the public sector's facility-based approach), the quality of usual care may have been greater than anticipated, leading to better outcomes. Additionally, the number of children included in the primary analysis was underpowered to detect a difference between the study arms. This loss of power occurred for two reasons. First, an ethics board interim review request after completing data collection led to use study-internal baseline z-scores for comparison and exclusion of some older children from the analysis. Second, the difficult rural geography and lengthy time-requirements for BSID-III assessments led to more caregivers than expected declining to complete follow-up assessments. Although there were few differences in baseline characteristics between subjects included versus excluded for these two reasons (see online supplementary tables 1–2), the proportion of children included in the analysis still remains only 43% of the originally randomised sample (figure 1). This potentially limits the generalisability of our findings, especially for the older children (aged 25–30 months at study exit), who represent the largest proportion of subjects excluded from the analysis.

Our study also has two important strengths. First, despite extensive work on stunting in rural indigenous Maya populations, there have been few contemporary efforts to investigate developmental outcomes in this population.[8–11] Our study represents, to our knowledge, the first report of the impact of a nutrition intervention on development in Maya infants. Second, despite its impact on overall study power, the use of study-internal z-scores for comparisons overcomes some concerns about the validity and reliability of the BSID-III for this indigenous population, allowing for robust internal comparisons, although the results we report here cannot be directly compared with populations from other studies.

Our study contributes to the literature on complementary feeding interventions, a cornerstone of stunting prevention and treatment efforts in low-income and middle-income countries.[13–15] Although multiple studies and meta-analyses have demonstrated the importance of complementary feeding interventions for promoting linear growth,[13–15] there are fewer studies examining their impact on development.[17 18 36] Here, we show that both usual care (lower intensity) and a higher-intensity approach to complementary feeding have important developmental effects. Our finding of no difference between higher-intensity and lower-intensity approaches contrasts with a large cluster-randomised trial in India, where a more-intensive approach showed improved developmental outcomes.[37] Given our study's low power, and the fact that complementary feeding indicators significantly improved in our larger trial,[24] we believe that the final role of individualised, intensive approaches to complementary feeding in our setting is not yet settled.

Our study has two important implications for child nutrition policy in Guatemala. Guatemala's indigenous Maya population has some of the poorest nutritional outcomes in the world and, despite a recent resurgence public interest in this problem,[38 39] chronic political and financial instability threatens sustained public and private commitments. Our study demonstrates, for the first time, the developmental impacts that such interventions can have for Maya children at risk and will help to advance this critical national conversation. Furthermore, our study shows that even low-intensity interventions modelled on existing public policy guidelines can be of great benefit.[40] Second, we demonstrate here that, despite the cultural and linguistic challenges of developmental assessments in this population, such evaluations are feasible and can show important developmental impact even when growth outcomes are equivocal.[24] We call for other nutrition researchers and programme implementers in Guatemala to more routinely incorporate developmental outcomes into their planned evaluations.

Future research priorities for our group include large-scale well-powered investigations of complementary and responsive feeding interventions, as well as more comprehensive, integrated nutrition and ECD interventions by CHWs in rural Guatemala. This latter point is especially important, since it is increasingly recognised that comprehensive, multisectoral interventions are most likely to generate sustained positive impact. Design and evaluation of comprehensive wrap-around interventions is also most in-line with the Nurturing Care Framework for childhood recently put forth by WHO and Unicef.[41 42] In addition, we are currently planning a re-enrolment study of this trial cohort to see if further growth or developmental benefits emerge or are sustained with longer follow-up, since an inherent weakness of this study was its short follow-up time. Finally, we plan to publish in greater

detail our methodology for adapting the BSID-III and related instruments to this population.

**Acknowledgements** The authors would like to thank the subjects, their families and communities for their participation in this study. The authors would also like to thank the interpreters at MHA and the graduate and undergraduate psychology students at Universidad del Valle de Guatemala for their collaboration performing the BSID-III assessments.

**Contributors** BM supervised the study, analysed and interpreted data and produced the first manuscript draft. SC cured, analysed and interpreted data. PR, ML, AG collected data and supervised BSID-III data collection. MFW and MPG supervised the study and cultural adaptation for the BSID-III. MPG and PR designed and supervised the study. No individual was given any form of payment to produce the article. All authors critically revised the manuscript and made substantial contributions to the final draft.

**Funding** This work was supported by Grand Challenges Canada, grant number SB-1726251050.

**Disclaimer** The study sponsor had no role in study design; the collection, analysis and interpretation of data; the writing of the report and the decision to submit the paper for publication.

**Competing interests** BM, MFW and PR are employees or volunteers at Wuqu' Kawoq, Maya Health Alliance, the non-governmental organisation that provided logistical support for this study in Guatemala. SC, PR, ML, AG and MPG are doctoral students or employees of Universidad del Valle de Guatemala. The authors declare no other relationships or activities that could have influenced the submitted work.

**Patient consent** Not required.

**Ethics approval** Institutional Review Boards of the Universidad del Valle de Guatemala and Maya Health Alliance.

**Provenance and peer review** Not commissioned; externally peer reviewed.

**Data sharing statement** Study protocol, replication dataset and statistical code are available on publication at: https://doi.org/10.7910/DVN/ATJ3NW.

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
