## [Reviewer comments · BMJ Paediatrics Open]

ARTICLE DETAILS

TITLE (PROVISIONAL)	Developmental outcomes of an individualized complementary feeding intervention for stunted children: A sub-study from a larger randomized controlled trial in Guatemala
AUTHORS	Martinez, Boris; Cardona, Sayra; Rodas, Patricia; Lubina, Meri; Gonzalez, Ana; Farley Webb, Meghan; Grazioso, Maria del Pilar; Rohloff, Peter

VERSION 1 – REVIEW

REVIEWER	Reviewer name: Sarah Nevitt Institution and Country: University of Liverpool, United Kingdom Competing interests: I have no competing interests
REVIEW RETURNED	04-Jun-2018

GENERAL COMMENTS	I have conducted a statistical review of the manuscript “Developmental outcomes of an individualized complementary feeding intervention for stunted children: A sub-study from a larger randomized controlled trial in Guatemala.” The authors report positive impacts on developmental outcomes in both the intervention and control groups, although no statistically significant difference between groups. Potential reasons for the lack of notable differences between groups are well described. Overall, this manuscript was interesting to read and methodology appears to be mostly appropriate. I have a few minor comments for the attention of the authors: 1) Introduction (Lines 101-103): “This is an important knowledge gap, since the political will for nutrition interventions in Guatemala is fragile, and more robust evidence of impact would have political significance.” This is a strong statement relating to politics – could a reference be added for this statement? Or if this statement represents the view of the authors, please clearly state this.2) Trial design (lines 133-134): It is not clear to me why only subjects recruited in the first five months of the original study were eligible for the sub-study (as opposed to all individuals in the larger study). Could the rationale for this inclusion criteria be clarified?3) Certainly, the use of the BSID-III scale seems to have been challenging and the study team have made extensive efforts to adapt the use of this scale to the study context and population in question (translation, visual aids, training etc.). While brief details of the adaptation of the tool are provided within the manuscript (sufficient detail for the purposes of the manuscript in my view); do the authors plan to write up in more detail their pilot study and methods for adapting the BSID-III tool for this study – perhaps in a similar format to references 3-6 of the protocol?
---

Such a methodological publication could be valuable for future research teams in similar cultural contexts

4) Sample size and randomisation (lines 170-171): "Allocation was by simple randomization from a computer-generated list of random numbers." I assume that allocation was performed within the original larger study - rather than any new randomization or allocation being performed for this sub-study?

5) Statistical analysis (lines 173-174): "Overall FCI scores were calculated as the sum of 18 item scores." I cannot find FCI scores reported within the results, am I missing something?

6) The method section (lines 181 to 186) describes the use of internal z scores at the request of an ethics board review. The results and discussion sections and Figure 1 then refer to exclusions of 'older children' following this ethic board request but it is not clear to me from the methods described why any exclusions had to be made under this approach.

7) The analysis approach is described as 'Intention-to-treat except where loss to follow-up prevented collecting outcome data (methods, line 176-177)' and "The analysis of primary outcomes was by intention-to-treat, including one child who discontinued treatment" (results, line 220-221). I am not disputing whether the approach taken is an intention-to-treat approach (according to the definition of intention to treat). However 'intention to treat' is commonly interpreted that all randomized individuals are included in analysis but here less than 50% of randomized individuals those who were eligible for the study were actually included in analysis (210 eligible and 100 included in analysis according to Figure 1).

While the reasons for the exclusions may be completely justifiable, I don't think that the extent of the exclusions are emphasised enough and the results could be misinterpreted.

8) Related to comment 5, very little information about the exclusions, including reasons for exclusions, is given in the results section. More details are given in the Discussion section (lines 265 to 272), including reference to Supplementary Tables which compare those included and excluded from analysis.

I suggest that more details of the excluded participants and the Supplementary Tables should be mentioned within the results, to provide context for the results of the included individuals. I also suggest noting within the text the differences in Age at BSID evaluation from Supplementary Table 2; 13.60 years in those included compared to 21.97 years for those excluded is quite a big difference.

9) Table 2 – this table shows the results for the subscales but for completeness, I suggest also adding the average scores to this table in a new row. These are important results which could potentially be missed if mentioned in the text only.

10) Discussion (lines 254-255): "These improvements remained highly statistically significant after controlling for important covariates (Table 3)." Perhaps I am missing something but I don't see where Table 3 shows the average change in the z-scores after adjustment for the important covariates.

	11) It is great that the authors have included the study protocol as a supplement to the manuscript. Please make reference to this supplement somewhere within the manuscript so that readers are aware it is there.
--	--

REVIEWER	Reviewer name: Raul Mercer Institution and Country: FLACSO (Latin American School of Social Sciences), Buenos Aires, Argentina (www.flacso.org.ar) Competing interests: No
REVIEW RETURNED	18-Jun-2018

GENERAL COMMENTS	The article is well and comprehensively written. Methodological issues dealing with the study design, inclusion criteria, statistical procedures, ethical considerations, were taken into account. Bibliography is exhaustive and updated. Minor changes and clarifications are requested as follows: The authors mentioned that there has been limited direct investigation of the effect of complementary feeding on developmental outcomes, Considering the present study design it becomes difficult to separate strictly those educational & feeding interventions from those oriented to promote exclusively ECD. The authors also mentioned that "this is an important knowledge gap, since the political will for nutrition interventions in Guatemala is fragile, and more robust evidence of impact would have political significance" Based on what evidence or reference the authors make this assumption? Particularly in Guatemala a country that historically has been a social experimental laboratory of nutritional research and interventions with poor results at the population level. Based on the lack of previous experience, what are your biggest concerns about the validity of the BSID-III applied to this indigenous population considering the cultural and language barriers? What is the explanation regarding the improvements in the cognitive subscale Z-scores that were more pronounced for males considering that there are studies with the opposite evidence? Why there is no explicit reference on the importance to have a broader and comprehensive approach to consider interventions considering family level and those that consider the structured social determinants of ECD and nutrition? On which bases were future research priorities been defined? Thank you.
--

REVIEWER	Reviewer name: Ana Garces, MD Institution and Country: INCAP: Guatemala Competing interests: None
REVIEW RETURNED	28-Jun-2018

GENERAL COMMENTS	Very interesting paper, on an innovative topic which has clear public health implications. With regards to the BSID, I would like to learn more about training procedures for the staff who performed the BSID, and also their background and how many evaluators were involved. Also, I think more detail needs to be incorporated into why the IRB requested use of different scales, as well as how and why this was done, as well as the implications of using a local scale to interpret the data.
--

VERSION 1 – AUTHOR RESPONSE

Reviewer #1:

1) Introduction (Lines 101-103): “This is an important knowledge gap, since the political will for nutrition interventions in Guatemala is fragile, and more robust evidence of impact would have political significance.” This is a strong statement relating to politics – could a reference be added for this statement? Or if this statement represents the view of the authors, please clearly state this.

We have qualified this statement, as follows: “This is an important knowledge gap, because feeding interventions may at times have small direct impacts on linear growth velocity but conceivably larger impact on developmental outcomes. Therefore, directly measuring these developmental outcomes may generate additional evidence supporting the interventions, which could have important policy implications in a country like Guatemala, where debate over the value of standard child nutrition program offerings is ongoing.” We have also added two additional citations to the reference list.

2) Trial design (lines 133-134): It is not clear to me why only subjects recruited in the first five months of the original study were eligible for the sub-study (as opposed to all individuals in the larger study). Could the rationale for this inclusion criteria be clarified?

We thank the reviewer for this comment. The principal reason for constructing a substudy of psychometric/developmental outcomes, rather than conducting this on the entire group of subjects from the larger study was that psychometric testing, especially in remote areas of Guatemala, involves very considerable time and cost expenditures. Now this section reads as follows: Due to the considerable time and expense constraints related to administering psychometric testing, a subset of subjects, namely those consecutively recruited for the larger study during the first five months (n=210, Figure 1), were invited to participate in the sub-study.

3) Certainly, the use of the BSID-III scale seems to have been challenging and the study team have made extensive efforts to adapt the use of this scale to the study context and population in question (translation, visual aids, training etc.). While brief details of the adaptation of the tool are provided within the manuscript (sufficient detail for the purposes of the manuscript in my view); do the authors plan to write up in more detail their pilot study and methods for adapting the BSID-III tool for this study – perhaps in a similar format to references 3-6 of the protocol? Such a methodological publication could be valuable for future research teams in similar cultural contexts

We thank the reviewer for this comment. We do plan to publish in more detail about the preparation and language and cultural adaptation of the BSID-III used in this study. We have added a final sentence to the “future directions” paragraph at the end of the conclusions: “Finally, we plan to publish in greater detail our methodology for adapting the BSID-III and related instruments to this population.”

4) Sample size and randomisation (lines 170-171): “Allocation was by simple randomization from a computer-generated list of random numbers.” I assume that allocation was performed within the original larger study - rather than any new randomization or allocation being performed for this sub-study?

We thank the reviewer for this observation. There was no further allocation or randomization for this sub-study. We have modified this section in the paper, now this section reads as follow: All subjects recruited for the larger study were allocated by simple randomization from a computer-generated list of random numbers, no further randomization or allocation were performed for this sub-study.

5) Statistical analysis (lines 173-174): "Overall FCI scores were calculated as the sum of 18 item scores." I cannot find FCI scores reported within the results, am I missing something?

Family Care Indicators scores are reported in Table 1 under Household Characteristics.

6) The method section (lines 181 to 186) describes the use of internal z scores at the request of an ethics board review. The results and discussion sections and Figure 1 then refer to exclusions of 'older children' following this ethic board request but it is not clear to me from the methods described why any exclusions had to be made under this approach.

We agree with the reviewer that this reasoning is complicated and we aren't certain we did the best job explaining it. Basically, we used baseline BSID-III scores from the entire cohort as they enrolled to calculate age-adjusted Z-scores. This allowed us to then derive Z scores at study exit using the reference set of score distributions generated at study entry for the entire cohort. However, since the cohort at enrollment was 6-24 months old, this means that those older children who, six months later were 25-30 months old and therefore we did not have any baseline score data to calculate Z scores for children in this age range.

The methods section has been modified and now reads as: Since all subjects recruited were younger than 24 months-old, baseline BSID-III Z-score data distribution was unavailable for subjects in older age bands at study exit, hence subjects older than 24 months-old were excluded from the primary outcome analysis.

7) The analysis approach is described as 'Intention-to-treat except where loss to follow-up prevented collecting outcome data (methods, line 176-177)' and "The analysis of primary outcomes was by intention-to-treat, including one child who discontinued treatment" (results, line 220-221). I am not disputing whether the approach taken is an intention-to-treat approach (according to the definition of intention to treat). However 'intention to treat' is commonly interpreted that all randomized individuals are included in analysis but here less than 50% of randomized individuals those who were eligible for the study were actually included in analysis (210 eligible and 100 included in analysis according to Figure 1).

While the reasons for the exclusions may be completely justifiable, I don't think that the extent of the exclusions are emphasised enough and the results could be misinterpreted.

We have added a few sentences to emphasize the exclusions. For example, the statistical analysis section has been modified and now reads as: Analysis was by intention-to-treat, except where loss to follow up prevented collecting outcome data or baseline age-band BSID-III data were unavailable to calculate exit BSID-III Z-scores. Under Outcomes, we now state, "The analysis of primary outcomes was by intention-to-treat for subjects younger than 24 months-old at study exit in this sub-study"

In our conclusions, in the paragraph on limitations, we have added another sentence, emphasizing the extent of the exclusions: "Although there were few differences in baseline characteristics between subjects included versus excluded for these two reasons (Supplementary Tables 1-2), the proportion of children included in the analysis still remains only 43% of the originally randomized sample (Figure 1). This potentially limits the generalizability of our findings, especially for the older children (aged 25-30 months at study exit) who represent the largest proportion of subjects excluded from the analysis."

8) Related to comment 5, very little information about the exclusions, including reasons for exclusions, is given in the results section. More details are given in the Discussion section (lines 265 to 272), including reference to Supplementary Tables which compare those included and excluded from analysis. I suggest that more details of the excluded participants and the Supplementary Tables should be mentioned within the results, to provide context for the results of the included individuals. I also suggest noting within the text the differences in Age at BSID evaluation from Supplementary Table 2; 13.60 years in those included compared to 21.97 years for those excluded is quite a big difference.

This section has been modified and now reads as: Subjects excluded were similar to those included, except for having lower minimum meal frequency (72% for excluded vs. 88% for included subjects) and being older (21.97 ± 1.71 vs. 13.60 ± 3.95 months-old). Difference in age was an expected finding, since our analysis was restricted to subjects younger than 24 months at study endpoint since baseline BSID-III data distribution for Z-scores calculation was unavailable for the older age bands.

9) Table 2 – this table shows the results for the subscales but for completeness, I suggest also adding the average scores to this table in a new row. These are important results which could potentially be missed if mentioned in the text only.

We thank the reviewer for this suggestion. We have added the Mean Change to Table 2.

10) Discussion (lines 254-255): “These improvements remained highly statistically significant after controlling for important covariates (Table 3).” Perhaps I am missing something but I don’t see where Table 3 shows the average change in the z-scores after adjustment for the important covariates.

In the original Table, this change is represented as “Time” and is the last row of the Table – (change in Z score from study entry to exit). This was probably not a very clear way of representing the change, so we have change the title of this row to “Adjusted Z-score change from study entry to exit” and moved it up to the top row of the table. We hope that this makes the table easier to read.

11) It is great that the authors have included the study protocol as a supplement to the manuscript. Please make reference to this supplement somewhere within the manuscript so that readers are aware it is there.

We have modified our Data Sharing Statement, where a copy of the study protocol will be stored for readers along with the replication data set and statistical code at the stable DOI

Reviewer: 2

The authors mentioned that there has been limited direct investigation of the effect of complementary feeding on developmental outcomes, Considering the present study design it becomes difficult to separate strictly those educational & feeding interventions from those oriented to promote exclusively ECD. The authors also mentioned that "this is an important knowledge gap, since the political will for nutrition interventions in Guatemala is fragile, and more robust evidence of impact would have political significance" Based on what evidence or reference the authors make this assumption? Particularly in Guatemala a country that historically has been a social experimental laboratory of nutritional research and interventions with poor results at the population level.

We have qualified this statement, and added additional references. The new sentence reads: “This is an important knowledge gap, because feeding interventions may at times have small direct impacts on linear growth velocity but conceivably larger impact on developmental outcomes. Therefore, directly measuring these developmental outcomes may generate additional evidence supporting the interventions, which could have important policy implications in a country like Guatemala, where debate over the value of standard child nutrition program offerings is ongoing.”

Based on the lack of previous experience, what are your biggest concerns about the validity of the BSID-III applied to this indigenous population considering the cultural and language barriers?

We think that the BSID-III, like most other psychometric tests are really designed for use in “WEIRD” populations (Western, educated, industrialized, rich, democratic), as they are often designated in the developmental literature. We don’t feel this precludes their use elsewhere, but they must be used with caution. Our use of Z scores derived internally from our study overcomes many of our concerns because, together with the randomized design, it allows for strong internal validity/internal comparisons within our study population. It does not however allow for comparison of our cohort’s scores with other population.

We added a few sentences to the “statistical analyses” section of the Methods to provide some additional detail, “BSID-III norms do not exist for Guatemala, and there are concerns about the validity of applying BSID-III norms from a different population. This is the case because, for example, some test items are most appropriate for urban, literate populations (e.g., use of books in the testing process) and because some language items require adaptation to the syntactic structure of Mayan languages”

What is the explanation regarding the improvements in the cognitive subscale Z-scores that were more pronounced for males considering that there are studies with the opposite evidence?

One likely explanation is that, in our cohort (and indeed, more generally in all of our work in rural Guatemala), boys are on average more stunted than girls. This is a finding that we and others have demonstrated repeatedly in most (but notably not all) cohort studies. We emphasized this study in the nutrition outcomes paper that we published on this study data (Martinez B, Webb MF, Gonzalez A, et al. Complementary feeding intervention on stunted Guatemalan children: a randomised controlled trial. *BMJ Paeds Open* 2018;2(1):e000213), but there and here also we are reluctant to speculate within the manuscript itself because we do not have any firm data. We suspect, however, that it is because of gendered differences in child rearing, and the fact that girls are more likely to be with mothers in the kitchen environment and therefore have more convenient access to foods. However, this requires careful study and, in fact one of our staff anthropologists is planning a detailed study to examine this gendered difference. Because all of this is highly speculative, we haven’t added any additional text to the manuscript, but could do so if the editor and reviewer feel it is important.

Why there is no explicit reference on the importance to have a broader and comprehensive approach to consider interventions considering family level and those that consider the structured social determinants of ECD and nutrition?

On which bases were future research priorities been defined?

These two points taken together from the reviewer are very important, and we agree that this broader framework is essential. We have modified the concluding paragraph of our manuscript to include some additional considerations and references along these lines: “Future research priorities for our group include larger-scale well-powered investigations of complementary and responsive feeding interventions, as well as more comprehensive, integrated nutrition and ECD interventions by CHWs in rural Guatemala. This latter point is especially important, since it is increasingly recognized that comprehensive, multisectorial interventions are most likely to generate sustained positive impact. Design and evaluation of comprehensive wrap-around interventions is also most in-line with the Nurturing Care Framework for childhood recently put forth by the WHO and UNICEF. [41,42]

Reviewer: 3

With regards to the BSID, I would like to learn more about training procedures for the staff who performed the BSID, and also their background and how many evaluators were involved.

More details in the background and training have been added to the manuscript. Now this section reads as follows: “Nine psychologists (bilingual Spanish/English; 3 postgraduate and 6 graduate or undergraduate) who had formal training in infant/child assessments as part of their education participated in a one week-long training in rural data collection and multicultural competencies, including active BSID-III practice sessions. Five Maya interpreters (Kaqchikel Maya/Spanish speaking; the majority with a high school teaching degree) participated in a one-week training with active BSID-III practice sessions. Senior study psychologist (MPG), with significant professional experience in infant/child assessments, supervised BSID-III training and data collection procedures. Testing was performed by a team of one psychologist and one interpreter, supervised by MPG and the principle study coordinator (BM).”

Also, I think more detail needs to be incorporated into why the IRB requested use of different scales, as well as how and why this was done, as well as the implications of using a local scale to interpret the data.

We thank the reviewer for this comment. For clarification, an IRB interim review requested the use of internal BSID-III scores distributions for comparison over the use of international norms. No additional scales other than the BSID-III were used in our study, just that we did not use norms from other populations (for example the USA) to standardize our data but only used study internal data with no external reference norm. Reviewer 1 also requested some clarifications around our BSID Z scores (see point 6 above under Reviewer 1).

The modified paragraph is as follows: “Raw scores for each BSID-III subscale (cognitive, receptive and expressive language, fine and gross motor, socioemotional) were calculated as the sum of assessment items performed. BSID-III norms do not exist for Guatemala, and there are concerns about the validity of applying BSID-III norms from a different population. BSID-III norms do not exist for Guatemala, and there are concerns about the validity of applying BSID-III norms from a different population. This is the case because, for example, some test items are most appropriate for urban, literate populations (e.g., use of books in the testing process) and because some language items require adaptation to the syntactic structure of Mayan languages.[31, 32] Furthermore, after data collection for the study was completed, interim ethics board review requested that we not use external norms in our analysis plan. Therefore study-internal Z-scores for each BSID-III subscale were calculated based on the raw scores distribution of each age-band during baseline measurements, an approach used in other studies.[33, 34] These were then used to calculate individual subject Z-scores for both 0 and 6 month timepoints. Since all subjects recruited were younger than 24 months-old, baseline BSID-III Z-score data distribution was unavailable for subjects in older age bands at study exit, hence subjects older than 24 months-old were excluded from the primary outcome analysis.”

In terms of implications of using internal baseline distributions for Z-scores calculations, in the conclusion section of the manuscript, we clarify: “Second, despite its impact on overall study power, the use of study-internal Z-scores for comparisons overcomes some concerns about the validity and reliability of the BSID-III for this indigenous population, allowing for robust internal comparisons, although the results we report here cannot be directly compared with populations from other studies..